# Electron Attachment Studies with the Potential Radiosensitizer 2-Nitrofuran

**DOI:** 10.3390/ijms21238906

**Published:** 2020-11-24

**Authors:** Muhammad Saqib, Eugene Arthur-Baidoo, Milan Ončák, Stephan Denifl

**Affiliations:** 1Institute for Ion Physics and Applied Physics, University of Innsbruck, Technikerstrasse 25, 6020 Innsbruck, Austria; Muhammad.Saqib@uibk.ac.at (M.S.); Eugene.Arthur-Baidoo@uibk.ac.at (E.A.-B.); 2Center for Biomolecular Sciences Innsbruck, University of Innsbruck, Technikerstrasse 25, A-6020 Innsbruck, Austria

**Keywords:** nitrofuran, reduction, radiosensitizer, fragmentation, electron attachment, low-energy electron

## Abstract

Nitrofurans belong to the class of drugs typically used as antibiotics or antimicrobials. The defining structural component is a furan ring with a nitro group attached. In the present investigation, electron attachment to 2-nitrofuran (C_4_H_3_NO_3_), which is considered as a potential radiosensitizer candidate for application in radiotherapy, has been studied in a crossed electron–molecular beams experiment. The present results indicate that low-energy electrons with kinetic energies of about 0–12 eV effectively decompose the molecule. In total, twelve fragment anions were detected within the detection limit of the apparatus, as well as the parent anion of 2-nitrofuran. One major resonance region of ≈0–5 eV is observed in which the most abundant anions NO_2_^−^, C_4_H_3_O^−^, and C_4_H_3_NO_3_^−^ are detected. The experimental results are supported by ab initio calculations of electronic states in the resulting anion, thermochemical thresholds, connectivity between electronic states of the anion, and reactivity analysis in the hot ground state.

## 1. Introduction

Together with chemotherapy and surgery, radiotherapy remains one of the most common and efficacious techniques for cancer treatment; globally, about 50% of patients receive radiotherapy as part of their treatment [1]. In radiotherapy, high-energy radiation is used to damage the genetic material (deoxyribonucleic acid, DNA) of tumor cells essentially via the formation of ions, radicals, and secondary low-energy electrons (LEEs) and thus to block their ability to divide and proliferate further [2,3]. At the molecular level, electrons even with insufficient energies for electron ionization can induce damage in biological molecules via a resonant process called dissociative electron attachment (DEA) [4]. DEA is initiated by the resonance capturing of a free electron resulting in the formation of an initial temporary negative ion (TNI) state. This TNI state is short-lived and may result in a subsequent dissociation of one or more bonds, leading to a negatively charged ion and neutral fragment(s). DEA may play a crucial role in the damage induced by ionizing radiation [5].

Radiosensitizers are a class of compounds that provide an additional contribution to target tumor cells by inhibiting cell proliferation and sensitizing hypoxic cells by mimicking the effects of oxygen [6]. Despite their profound advantages in a hypoxic environment, the most important question is how to obtain a balance between their efficiency and toxicity, which is a key factor that has led to the failure of many radiosensitizers clinically [7]. Therefore, the molecular mechanisms underlying these advantages are still being investigated. The compounds containing nitric oxide (NO) and/or a nitro functional group (NO_2_) turned out to be highly effective, even transcending the effect of oxygen itself [8,9,10]. The NO_2_ group is investigated to be an adaptable and exclusive functional group in medicinal chemistry. Currently, several articles report that the major work of medicinal chemists in the last years has been focused on the exploration of potential anticancer molecules containing NO_2_. These compounds persuade their damaging effects on cells by different processes, such as topoisomerase inhibition [11,12] or DNA alkylation [13], and have significantly demonstrated hypoxia-induced effects that are attributed to their bioreductive activation potential [13,14,15]. Among them, nitroimidazole turned out to be a useful heterocyclic moiety that has been extensively employed to exert diverse biological effects and has often been explored as a bioreductive arm [10,16]. Nitro derivatives have very prominent electron acceptor properties due to the strong electron-withdrawing inductive effect of the nitro group and due to the low energy of the lowest unoccupied (π*) orbital [17]. These beneficial electron acceptor properties have been also observed in electron attachment studies with nitroimidazolic compounds such as nimorazole [18], misonidazole [19], and isomers of nitroimidazole [20]. For this reason, the knowledge about the response of a radiosensitizer upon low-energy electron interaction is important to define its characteristic behavior. 

Furan derivatives, especially nitrofurans and benzofuran, are well known for their antimicrobial potential [21,22]. The furan ring has gained exceptional attention due to its crucial performance in combustion chemistry, biomass pyrolysis, and due to some interesting aspects of its thermochemistry, especially to its very high C–H bond strength [23,24,25]. Since the last century, nitrofurans have been used in human and veterinary medicine, mainly for the cure of bacterial and protozoan infections. Their functional principle has been ascribed to the dielectronic reduction of the nitro group by nitro reductases from microorganisms, generating the electrophilic intermediaries nitroso and hydroxylamine, which in turn can bind to DNA and other biomolecules in microorganisms [22,26]. Nitrofuran compounds were suggested as potential radiosensitizers based on the calculation of their electron affinity [27]. In the current study, we explore negative ion formation upon electron attachment to 2-nitrofuran, 2-NIF (see Figure 1 for the structure of this cyclic aromatic compound, in which NO_2_ binds to the furan ring at the C2 position) by measuring the anion efficiency curves at high electron energy resolution. DEA to furan was studied formerly [28,29]. Five anionic species were reported by Sulzer et al. [28]. They observed the formation of the three most abundant ions, C_2_HO^−^, C_4_H_3_O^−^, and C_3_H_3_^−^, at the electron energy around ≈6 eV. At these energies, the capture process is often ascribed to core-excited resonances, where an outer-valence electron of the molecule is promoted to a formerly unoccupied orbital. In the early 1990s, Muftakhov et al. [29] reported 10 anionic fragments formed around 5–7 eV upon DEA to furan. They suggested that anionic fragments were also formed via an electronically exited intershell resonance. Although electron transmission spectroscopy with the furan molecule showed low-lying TNI states below the electronic excitation threshold [30], the absence of low-energy DEA ion yield is in line with the high bond dissociation energies for hydrogen loss [25]. Studying DEA to 2-NIF in an electron energy range from 0 to 12 eV, we have found low-energy DEA resonances for 2-NIF, where due to the relatively high C–H bond dissociation energy, the formation of an NO_2_ radical has prevailed over OH radical formation. Such characteristics of 2-NIF may be relevant toward radiosensitizing capabilities upon the interaction with secondary electrons. To the best of our knowledge, there is no information and data available regarding the energetics of the electron attachment to 2-NIF.

## 2. Methods

### 2.1. Experimental Setup

The current study of electron attachment was carried out using a crossed electron–molecular beam instrument. More details about the setup can be found in Ref. [31]. The instrument includes an oven with a capillary of 1 mm inner diameter to introduce the sample inside the chamber as a molecular beam, a hemispherical electron monochromator (HEM) that serves as an ionization source, a quadrupole mass analyzer, and a channel electron multiplier with a pulse-counting system. The experiment is operated in a high vacuum (≈10^−8^ mbar background pressure). Before starting the measurements, different inlets have been tested for the formation of an effusive molecular beam. Initially, the crystalline 2-NIF sample was placed in the oven inside the vacuum chamber. However, it turned out that 2-NIF has a high vapor pressure and readily evaporates at room temperature. Therefore, an external sample container was used instead of the heated oven. The sample container was combined with a 1 mm diameter capillary tube (made of stainless steel) via an inlet valve system, which allowed the more controlled introduction of the sample. The working pressure in the vacuum chamber was 2.7 × 10^−7^ mbar. The formed effusive beam entered the interaction region of the HEM. The HEM comprises three parts: a hairpin tungsten filament (heated by applying a current of ≈2.35 A) used as the electron source, two concentric hemispheres at different electric potential, which function as an energy selector, and two columns made of a sequence of electrostatic lenses. The latter component is used to guide the electron beam from source to hemisphere and from hemisphere to interaction region. The electron beam, with 15–20 nA current, was directed to the interaction region by the HEM with an energy resolution of 100 meV full width half maximum (FWHM). The resolution was determined by the FWHM of the well-known sharp 0 eV resonance of Cl^−^ formed upon DEA to CCl_4_ [32]. Before the measurement of anions from 2-NIF, energy calibration was performed. A Faraday cup, connected with a picoampere meter, was placed right after the interaction region to monitor the amount and stability of the electron beam. Subsequently, the anions created in the interaction region were extracted by a weak electric field and directed to a quadrupole mass analyzer with a nominal mass range of 2048 u. Finally, the mass-analyzed ions were detected by a channeltron and recorded by a preamplifier with an analog-to-digital converter unit. For recording a mass spectrum, the electron energy was kept constant, and the ion yield was recorded as a function of the mass. Whereas for the anion efficiency curve of a given ion, the corresponding mass was set with the quadrupole, and the electron energy was varied in a certain range. We also determined the electron attachment cross-section for the 2-NIF parent anion and the NO_2_^−^ fragment anion by comparing the obtained signals for 2-NIF with the well-known cross-section of the 0.8 eV peak of Cl^−^ anion of CCl_4_ [33]. Prior to that, the measured signals for 2-NIF and CCl_4_ were normalized by dividing the value by the corresponding partial pressure of the compound.

### 2.2. Computational Methods 

The nitrofuran molecule and its fragments were optimized in the ground electronic state of both neutral and anionic molecules at the B3LYP level and recalculated as single points using the Coupled Cluster Singles and Doubles method, CCSD, with the aug-cc-pVTZ basis set. The electron affinity predicted at the B3LYP level is by 0.2–0.3 eV higher compared to the CCSD single-point recalculation, with negligible basis set effects. Benchmark calculations on related maleic anhydrate show that the CCSD value for adiabatic electronic affinity (1.34 eV) is closer to the experiment (1.44 ± 0.09 eV) [34] compared to the B3LYP value (1.71 eV). Therefore, we use the CCSD values below.

Excited states of the nitrofuran anion were calculated using the time-dependent density functional (TDDFT) approach, namely TD-CAM-B3LYP, and the Equation of Motion–CCSD (EOM-CCSD) method; up to 25 excited states were considered. Among the excited states, all excitations into diffuse orbitals were ignored with the exception of the first transition that represents the dipole-bound state (DBS); excitations from inner orbitals into the single occupied molecular orbital (SOMO) are also ignored, as they do not represent states that can be reached directly through electron attachment. Using this simplified analysis, we ignore more complex electron attachment mechanisms, e.g., Feshbach resonances. For a description of the DBS, the aug-cc-pVTZ basis set was extended by *s* and *p* Gaussian functions with the exponent of 0.01 placed on hydrogen atoms. Target orbitals into which an electron is placed were calculated by natural orbital analysis [35] of the excited states of the anionic species.

The description of chemistry following electron attachment represents a challenge, as many electronic states with diffuse character are present, rendering calculations with a larger basis set complicated. In the literature, both multi-reference [36] and EOM-CC-based approaches [37,38] were employed to model electron attachment. Here, we used the Complete Active Space–Self Consistent Field (CASSCF) and Multireference Configuration Interaction (MRCI) methods, along with a limited active space of 3 electrons in 6 orbitals, denoted as (3,6), to cover the most important valence states (see Appendix A for included orbitals). The 6–31g* basis set was used, because such a small basis set enabled us to limit ourselves to only few representative electronic states; already with 6–31+g*, several electronic states of diffuse character appear, with the need to increase the size of active space and the number of included electronic states. Although this limitation might influence results quantitatively, we use it to approach the qualitative features of the relaxation to the ground electronic state after electron attachment. To speed up the calculations, 13 orbitals were cored during the MRCI procedure. Averaging over four lowest electronic states was employed.

All reported reaction energies include zero-point correction calculated with the B3LYP method. CASSCF/MRCI calculations were performed in the Molpro program [39], and all other calculations were done in the Gaussian software package [40].

## 3. Results and Discussion

As described by the following reaction equations, electron attachment to a neutral molecule M leads to the formation of the TNI M*^−^ and can be described as a vertical transition from the potential energy curve of the neutral to the ionic state, following thereby the Franck–Condon principle. Afterwards, the TNI either relaxes by fragmentation into the anionic fragment A and the neutral fragment(s) B, known as DEA process Equation (1) or becomes neutral again by auto-detachment Equation (2):(1)$$e^{-}+\;M\;\to \;M^{*-}\;\to \;\;A^{-}+B$$
(2)$$e^{-}+\;M\;\to \;M^{*-}\;\to \;\;M+e^{-}$$

Initially, we recorded a series of negative ion mass spectra at different electron energies to discern the fragmentation pattern of 2-NIF upon electron attachment. Figure 1 shows the ion yield as a function of the mass-to-charge ratio (*m*/*z*) in a summed mass spectrum. This spectrum was obtained by the sum of 11 individual spectra in the 0–10 eV range (recorded in steps of 1 eV). The found mass peaks were assigned to negatively charged reaction products according to their mass and quantum chemical calculations; see also Table 1. Thirteen anionic species have been detected in the studied electron energy range, including a metastable parent anion at *m*/*z* 113. Hence, the sensitivity of 2-NIF toward low-energy electrons is of particular interest. The highest peak at *m*/*z* 46 is attributed to the fragment anion NO_2_^−^, with almost twice the signal of the parent anion. The formation of the complementary anion C_4_H_3_O^−^ was much weaker; other fragment anions are formed with similar intensities. In the original spectrum (measured with the optimum transmission of anions through the quadrupole at limited mass resolution), it was not possible to resolve mass peaks in the range of *m*/*z* 25–28. An improved measurement is shown as an inset in Figure 1, and it resolves consecutive anionic fragments of similar intensity at *m*/*z* 25, 26, and 27.

Subsequently, the anion efficiency curves (anion yield as a function of electron energy) of the anions identified in the mass spectra have been measured and are shown in Figure 2 and Figure 3. The energies of peak maxima X_max_ reported in Table 1 were determined by Gaussian fits. The experimental threshold for each anion (also listed in Table 1) was determined by a method proposed by Meißner et al. [41] as X_onset_ = X_max_ (of first resonance) − 2*σ*, where *σ* is the standard deviation of the peak. We also determined the total anion yield as a function of the electron energy by summing up all the recorded anion efficiency curves. Such an experimentally obtained total ion yield is shown in Figure 4 along with the positions of calculated states and target orbitals in the respective anion. The present results show that low-energy electrons are very efficiently trapped by 2-NIF, with most anionic fragments formed in the energy range of 1–5 eV. Our calculations predict for 2-NIF the vertical electron affinity of 0.60 eV, the adiabatic one of 0.98 eV (CCSD/aug-cc-pVTZ//B3LYP/aug-cc-pVTZ). In the ground electronic state of the resulting anion, D_0_, the electron is delocalized over the whole molecule, and the bonds in the furan ring are weakened. We also found an excited state at the electron energy of 0 eV, which can be assigned to a dipole bound state with the electron positioned next to hydrogen atoms (the dipole moment of 2-nitrofuran is calculated as 5.3 Debye at the B3LYP/aug-cc-pVTZ level).

Figure 4 shows that the calculated position of other excited states also matches well to the experimental data. The electron attachment at ≈1.5 eV might populate orbitals that are more localized on the ring than on the NO_2_ moiety, with a considerably antibonding character along the C–N axis, hinting toward a possible NO_2_^0/−^ formation. Four different resonances are predicted for the broad peak close to 3 eV, again with a partial antibonding character for the C–N bond.

In the energy range close to 0 eV, corresponding to the initially formed dipole-bound state, only the molecular anion, 2-NIF^−^, is observed (see Figure 2a). The intact parent anion is the second most abundant anion observed, with an estimated cross-section of 2.3 × 10^−20^ m^2^. The absence of decomposition reactions in the low-energy region agrees with the calculated energetics of dissociation reactions (Table 1). All reactions are endothermic except for the CN^−^ channel, which requires considerably molecular restructuring as discussed below. In other words, there is no open dissociation channel for anions created at ≈0 eV. The formation of an intact parent ion by the attachment of a free electron in the gas phase following collision-free conditions dictates that the excess energy stored by the attachment, which generally consists of the kinetic energy of the interacting electron (which is negligible in the present case, since the parent anion is only formed at ≈0 eV) and the electron affinity of the molecule, is efficiently distributed across the vibrational degrees of freedom in the TNI [42]. Accordingly, auto detachment is postponed, and the resulting lifetime extends to the µs time regime, which enables us to observe the parent anion by mass spectrometry. In electron attachment experiments with furan (C_4_H_4_O), no parent anion was reported, which was in accordance with the prediction of a negative electron affinity (–1.76 eV) [43]. In this case, the anionic state is energetically above the ground state of the neutral and hence prone to autodetachment or DEA before experimental detection by mass spectrometry. Similar to 2-NIF, the addition of a nitrogen dioxide group also had a stabilizing effect on the formation of the parent anion for other small aromatic molecules such as benzene [44] and uracil [45]. Interestingly, for nitroimidazole isomers, no parent anion was observable by mass spectrometry [20,46,47]. However, the observation became possible by an additional replacement of the hydrogen at N1 by a methyl group [46,47]. Such a pronounced effect can be explained by the complex competition between the intramolecular redistribution of excess energy, DEA, and autodetachment. While high (energy) resolution DEA studies with nitrobenzene indicated that all DEA channels are energetically closed at zero eV [44] and hence parent anion formation only competes with autodetachment, few exothermic DEA channels are energetically open for nitroimidazole. Upon methylation, these DEA channels are blocked, and only the competition with autodetachment remains as in the case of nitrobenzene. Larger nitroaromatic compounds such as the potential radiosensitisers nimorazole and misonidazole offer a larger number of vibrational degrees of freedom, which allows detection of the parent anion [18,19].

In addition to the stable parent anion, we observed several fragment anions formed by DEA reactions in the current study. The cleavage of the C2-NO_2_ bond upon DEA leads to the formation of the NO_2_^−^ fragment anion (mass 46 u). The corresponding anion efficiency curve is shown in Figure 2b. As mentioned above, there is a clear preference of excess charge localization at the nitro group upon cleavage of the C2-NO_2_ bond, which is underlined by the anion efficiency curve of the far less abundant C_4_H_3_O^−^ (67 u) displayed in Figure 2c. This experimental result agrees with the calculations that predict that electron localization on NO_2_ is by about 0.5 eV more favorable than on the ring fragment. The estimated DEA cross-section for NO_2_^−^ formation is 7.8 × 10^−20^ m^2^, which is similar to that obtained for the nitrobenzene molecule (4.6 × 10^−20^ m^2^) [44]. These values are about one order of magnitude lower than those of the nimorazole compound, which is composed of the nitroimidazolic moiety with an attached morpholine ring. Taking into account the stated uncertainty of one order of magnitude [18], the cross-section values indicate that DEA is also an efficient process for small aromatic compounds.

The anion efficiency curves of NO_2_^−^ shows the main resonance at 3.1 eV, while a weaker resonance appears at 1.5 eV. We note that similar resonance positions in the NO_2_^−^ ion yield were reported for nitroimidazole [46]. Kossoski and Varella computationally investigated these resonances and proposed an indirect dissociation process for NO_2_^−^ formation, in which the coupling of π* states and the repulsive σ*_CN_ states occurs [48]. For nimorazole [18] and misonidazole [19], the same peak shape was observed in the NO_2_^−^ anion efficiency curve; however, an additional peak structure very close to the electron energy of ≈0 eV was present with strongly varying intensity.

The complementary anion of NO_2_^−^, C_4_H_3_O^−^, is also observed at 3.1 eV (see Figure 2c), with no clearly discernible feature close to 1.5 eV present in the anion yield. The shape of the anion yield measured at *m*/*z* 55 is almost identical to the NO_2_^−^ anion yield, as shown in Figure 2d. We ascribe the ion yield to the C_3_H_3_O^−^ fragment anion. Formation of the alternative C_2_HNO^−^ ion with the same nominal mass seems to be more complicated from the kinetic perspective and was calculated to lie higher in energy.

The anion efficiency curves as a function of the incident electron energy for the fragment anions with masses between 16 and 42 u are shown in Figure 3, with the main peak found in the range of 3.3–4.3 eV for all ions, i.e., slightly higher than for NO_2_^−^ with its maximum at 3.1 eV. Based on thermochemical calculations and kinetic considerations, ions in the range of 25–42 u are assigned to further decomposition products of C_4_H_3_O^−^ and C_3_H_3_O^−^, with calculated reaction energy matching well to the position of experimental peaks. The fragment anion C_2_H^−^ is a known stable compound with an electron binding energy of 2.97 eV [28,49] and was among the fragment anions reported in DEA studies to furan [28]. The ion yield obtained at *m*/*z* 26 could correspond to both C_2_H_2_^−^ (a renowned anion having the vinylidene structure C=CH_2_ [47,50]) and CN^−^; calculations suggest that both ions could be present in the experiment, the low experimental yield of the strongly exothermic CN^−^ channel can be explained by necessary bond reorganization (see below). The signal obtained at *m*/*z* 27 may be nominally assigned to C_2_H_3_^−^ or HCN^−^. However, a previous study [51] reported negative electron affinities of HCN^−^ and NHC^−^, which exclude their observation on mass spectrometric timescales.

Finally, the formation of the O^−^ anion may result by single bond cleavage from the nitrogen dioxide group and was therefore not observed in DEA to furan [29]. In DEA to 2-nitroimidazole, the O^−^ anion was not observed; [20] instead, the smallest anion detected was OH^−^, which is not observed here. The ion yield shape is similar to those for the other small fragment anions and just shifted by few hundreds of meV toward higher electron energies. This difference in fragmentation may be related to the availability of a readily detaching hydrogen upon electron attachment to the compound, which influences the chemistry induced by low-energy attachment. For 2-nitromidazole, such hydrogen is available at N1 position, which also led to the very abundant formation of (M–OH)^−^. We investigated whether the loss of neutral hydroxyl radical would be an abundant channel and measured the ion yield of (2-NIF–OH)^−^ at *m*/*z* 96. The corresponding ion yield is shown in Appendix A and is very low compared to other fragment anions. In the case of furan, the bond dissociation energies for the C–H bond are known to be high [25], and therefore, OH dissociation may become a less favorable process in contrast to nitroimidazoles. We also measured the anion yield of (2-NIF–O)^−^ at *m/z* 97 (see Appendix A). The ion yield is within the same order of magnitude as (2-NIF–OH)^−^, which indicates that preferred localization of the excess charge at the oxygen atom occurs.

After an electron is attached to the nitrofuran molecule, the system might funnel to the ground electronic state of the anion. A possible pathway is analyzed in Figure 5 using multi-reference calculations with a limited 6–31g* basis set in order to account only for the most important electronic states of the resulting anion. Namely, the anionic ground state and two electronically excited states denoted as **2** and **4** in Figure 4 are included; on the other hand, the DBS and other electronic states are missing from the state average. The pathway shows that following the electron attachment measured experimentally at ≈1.5 eV, the system relaxes into a minimum on the excited state potential surface characterized by the broken planarity of the molecule, with hydrogen atoms above and below the ring. However, it may reach a conical intersection, i.e., a crossing with the electronic ground state, with the NO_2_ group in a pyramidal configuration. Along the pathway, the NO_2_ group rotates with respect to the ring. No excess energy is needed, and therefore, this pathway is available for the system readily after electron attachment in the band at 1.5 eV corresponding to the state **2**. From the conical intersection, the system can reform a structure close to the Franck–Condon (FC) point and follow dissociation pathways in the hot ground state. For higher-lying electronic states within the 1.5–4 eV band, we expect that they reach the state **2** through other conical intersections and follow the same pathway into the ground electronic state.

By inspecting the dissociation dynamics, we also further analyzed the mechanisms of how the experimentally observed anions emerge. In the ground electronic state of 2-NIF^−^, several dissociation channels are available for the molecule; the respective potential energy surface is shown in Figure 6. On the left-hand side, direct dissociation channels are shown—both ions formed by C–N dissociation, O^−^ formation observed for high energies, and H dissociation not observed in the present experiment. On the right-hand side, a pathway to C_3_H_3_O^−^ is shown. The formation of this ion by the evaporation of CO and NO is the second lowest-energy channel among the reactions listed in Table 1. However, it requires considerable molecular rearrangement that starts with breaking the weakest bond of 2-NIF^−^ in the ground electronic state, the C–O bond in the furan ring (as also observed in a short molecular dynamics run at elevated temperature). When the ring is open, an oxygen atom might migrate over the molecule through an energy-demanding transition state, forming a very stable ONCOC_3_H_3_O^−^ fragment. There is a considerable uncertainty in the energy of the transition state, with B3LYP energetics favoring C_3_H_3_O^−^ over C_4_H_3_O^−^, which is in agreement with the experiment; optimization at the CCSD level would probably shift the transition state energy to lower values. From the rearranged structure, the evaporation of NO and CO molecules might take place, with the energy needed for this dissociation well below the height of the transition state that had to be surpassed. Then, the formed C_3_H_3_O^−^ anion may dissociate further, creating four out of six ions in Figure 3; the C_2_H_2_O^−^ ion is probably formed by the dissociation of NO and C_2_HO, sharing however the pathway to ONCOC_3_H_3_O^−^; the CN^−^ ion might be formed before NO and CO dissociation takes place. O^−^ is predicted to form through direct dissociation.

Figure 6 comprises all the main dissociation channels observed experimentally. Based on the shown energetics, one can qualitatively understand the reactivity of 2-NIF after electron attachment, provided that the system funnels down into the ground electronic state beforehand (Figure 5). As already mentioned above, for electron energies below 0.5 eV, only the C_3_H_3_O^−^ and CN^−^ channels can be reached thermochemically, but they are hindered kinetically. Therefore, the parent ion is observed in the experiment at low energies. At about 0.8–1.0 eV, the NO_2_^−^ channel opens and prevails also for higher energies due to its low activation barrier. Between 1.1 and 1.8 eV, the C_4_H_3_O^−^ and C_3_H_3_O^−^ channels open in the experiment. C_4_H_3_O^−^ origins through direct dissociation along the C–N coordinate; its curve in Figure 2 copies the one of NO_2_^−^, just with a cut-off for lower energies. For the C_3_H_3_O^−^ channel, the higher-lying barrier has to be overcome. For even higher energies, we see a further dissociation of C_3_H_3_O^−^ to smaller molecules or, possibly, C_2_H_2_O^−^ formation by the dissociation of NO and C_2_HO (see Table 1). The direct evaporation of O^−^ is favored energetically over hydrogen atom dissociation and is seen only at higher electron energies. All discussed features are consistent with dissociation patterns in the 0–5 eV region comprised in Figure 2 and Figure 3.

Finally, it should be noted that we obtain only three anions at electron energies above 6 eV in DEA to 2-NIF (see Figure 3). The corresponding maximum of the weakly abundant peak is found at 8.7 eV (C_2_H_2_O^−^/CNO^−^, C_2_HO^−^) and 9.1 eV (O^−^), respectively. We may ascribe such high-lying peaks to Rydberg excitation processes as previously suggested for DEA anion yields of furan in the electron energy range between 10 and 11 eV [28].

## 4. Conclusions

Different modes of action of radiosensitizers for hypoxic tumor tissues, which mimic the effect of oxygen [52], were suggested previously. For the nitroimidazolic compound nimorazole, it was proposed that its action is based on the initial reduction by low-energy electrons in tumor cells leading to an intact parent radical anion which (after protonation) represents the damaging species itself [18]. Such a mode of action is distinctly different from other radiosensitisers such as tirapazamine, which efficiently breaks up upon low-energy electron attachment at ≈0 eV and forms damaging neutral radical species such as the hydroxyl radical [53]. The present results show that dissociation reactions leading to the formation of neutral radicals are minor upon electron attachment to 2-NIF, and thus, the properties are more similar to nimorazole. This result may be also ascribed to the absence of open DEA channels for low-energy electrons close to 0 eV. The present calculations further show that for the least endothermic reaction channel leading to C_3_H_3_O^−^, a significant barrier of 1.79 eV exists. Notable is also the absence of the OH^−^ fragment anion, which was observed in DEA studies with the radiosensitiser misonidazole [19]. It was predicted that the formation of OH^−^ remains upon solvation [54], in contrast to NO_2_^−^, which was observed as an abundant fragment anion only under isolated conditions in the gas phase.

## Figures and Tables

**Figure 1 ijms-21-08906-f001:**
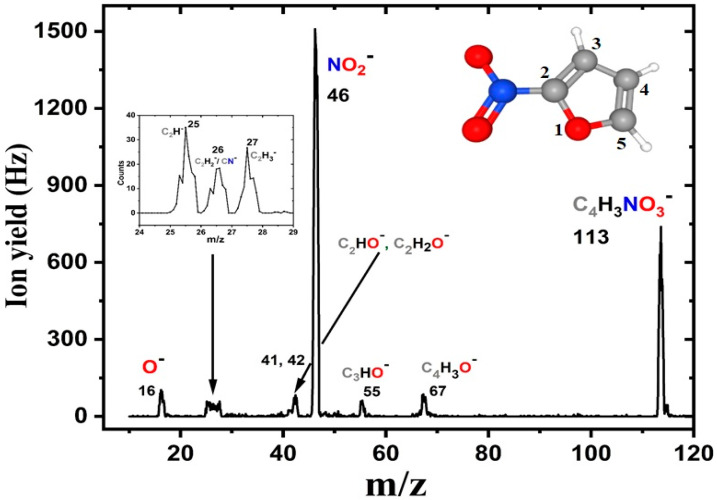
Negative ion mass spectrum of 2-nitrofuran upon electron attachment. The spectrum was obtained by summing 11 individual spectra recorded at electron energies from 0 to 10 eV, in steps of 1 eV. The inset shows anionic fragments formed in the range of *m/z* 25–28 with improved mass resolution. A schematic representation of the molecule is also shown.

**Figure 2 ijms-21-08906-f002:**
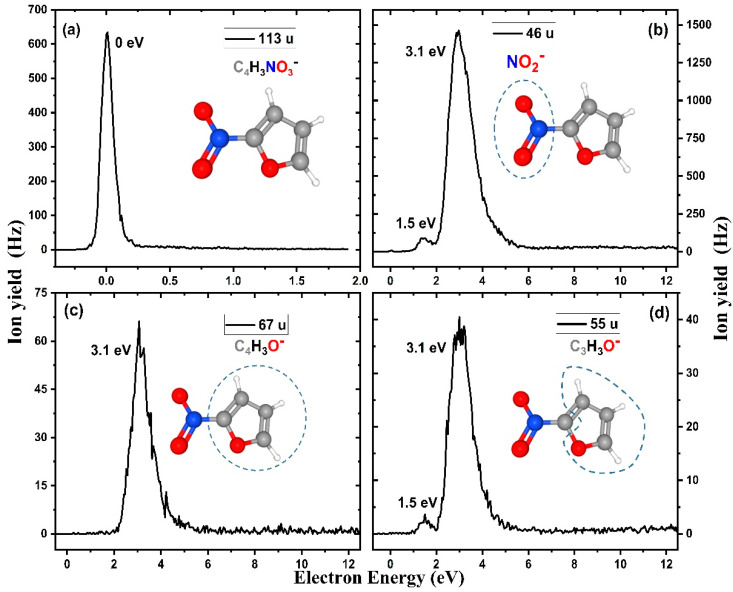
Anion efficiency curves of the four most abundant anions observed in the present electron attachment study with 2-nitrofuran (2-NIF), (**a**) the parent anion C_4_H_3_NO_3_^−^, (**b**) NO_2_^−^, (**c**) C_4_H_3_O^−^ and (**d**) C_3_H_3_O^−^, respectively.

**Figure 3 ijms-21-08906-f003:**
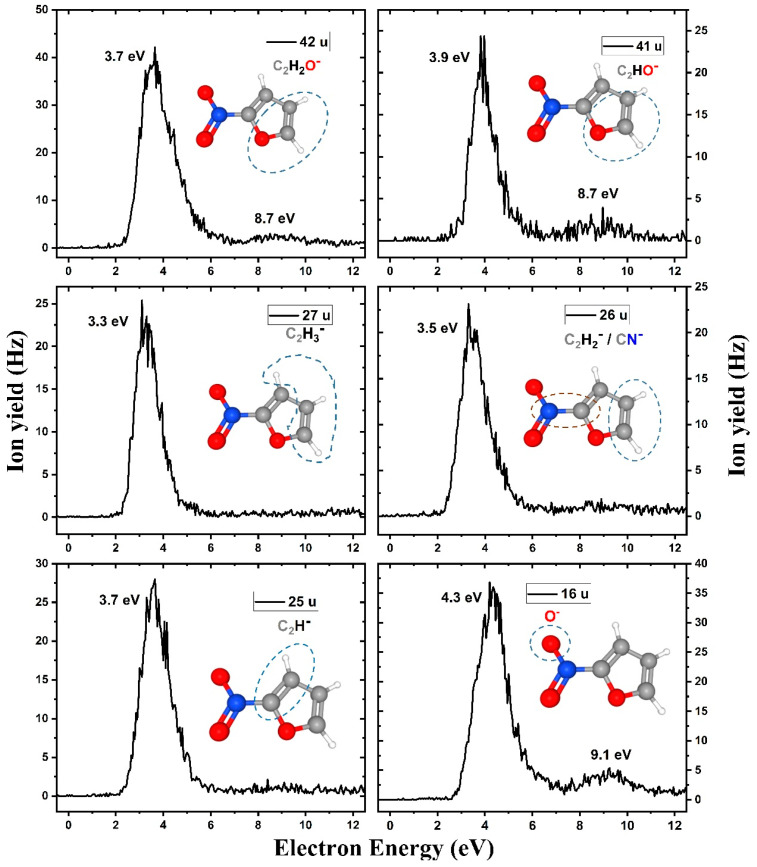
Anion efficiency curves for the anionic fragments with masses between 16 and 42 u formed upon dissociative electron attachment (DEA) to 2-NIF.

**Figure 4 ijms-21-08906-f004:**
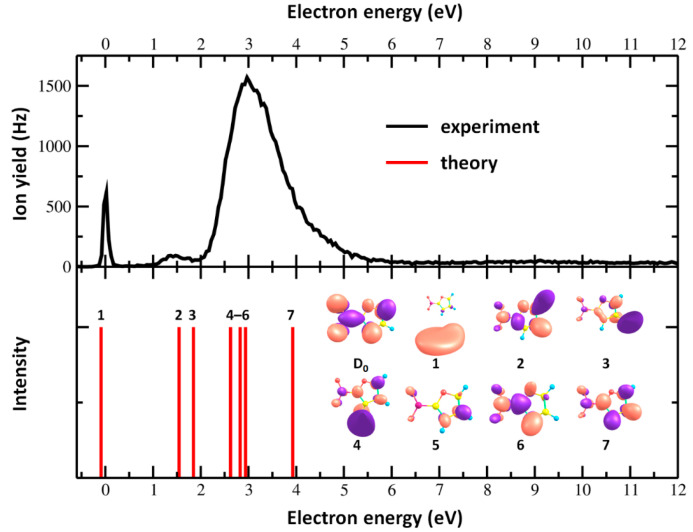
Total ion yield of measured anions formed upon electron attachment to 2-NIF (upper panel) and calculated position of resonances along with target orbitals for each calculated state (lower panel). The total ion yield was obtained by the sum of the mass-selected anion yields shown in Figure 2 and Figure 3. The excited states were calculated at the TD-CAM-B3LYP/aug-cc-pVTZ//B3LYP/aug-cc-pVTZ level of theory. Twenty-five excited states were considered, with transition probability put equal for every transition. Apart from the dipole bound state, only non-diffuse states and states that can be created by direct electron attachment to the neutral molecule were considered; therefore, higher electronic states are not described as D_x_, and only the D_0_ state is explicitly denoted. Calculated transition energies for the target anionic states are shifted by –0.60 eV (negatively taken vertical electron affinity calculated at the CCSD/aug-cc-pVTZ//B3LYP/aug-cc-pVDZ level) and –0.18 eV (average difference of excitation energies between TD-CAM-B3LYP and EOM-CCSD with the aug-cc-pVDZ basis set for first 25 excited states), i.e., all states are shifted by –0.78 eV. For a description of the dipole-bound state, additional basis set functions were employed (see Section 2).

**Figure 5 ijms-21-08906-f005:**
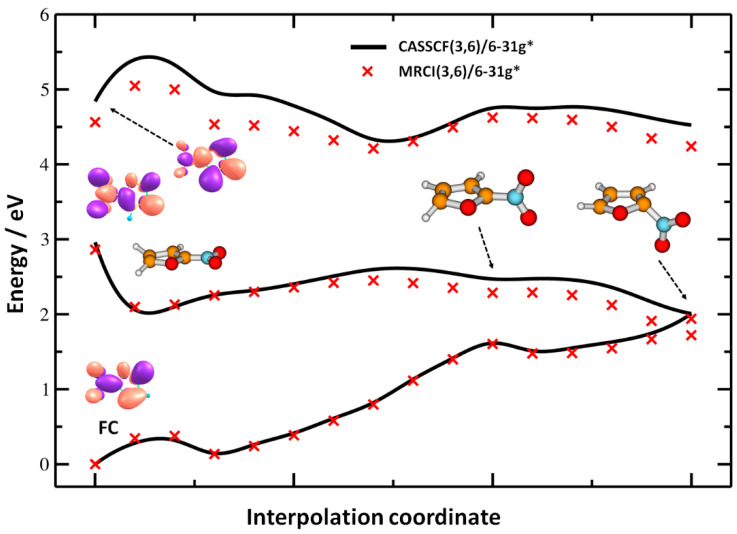
Reaction pathways in the electronically excited nitrofuran anion optimized at the CASSCF(3,6)/6–31g* level and single-point recalculated at the MRCI(3,6)/6–31g* level (black lines and red crosses, respectively). The Franck–Condon (FC) point (first point), the minimum on the excited state potential energy surface (second point), constrained optimization in the D_1_ state with respect to the NO_2_ group rotation (points 3–11) and interpolation to a conical intersection with pyramidalized NO_2_ group (points 12–16) are shown. Orbitals corresponding to excitations in the FC point were calculated at the CASSCF level. Note that due to the small basis set used, not all states depicted in Figure 4 could be reproduced here (see Section 2 for details).

**Figure 6 ijms-21-08906-f006:**
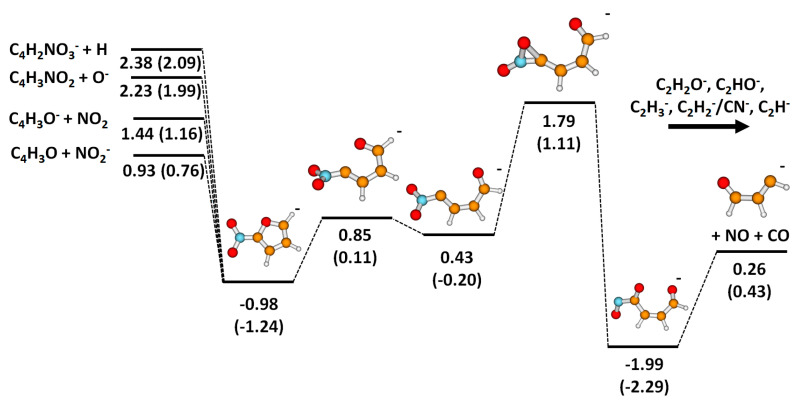
Potential energy surface mapping dissociation channels in 2-NIF^−^. Calculated at the CCSD/aug-cc-pVTZ//B3LYP/aug-cc-pVTZ level, the respective B3LYP values are provided in parentheses. The energy is given in eV.

**Table 1 ijms-21-08906-t001:** Identification of reaction channels according to measured mass, calculated reaction energies ΔE_theo_ at the CCSD/aug-cc-pVTZ//B3LYP/aug-cc-pVTZ level, experimental energy onset X_onset_ defined as X_max_ − 2σ and resonance position X_max_ (all in eV). The dehydrogenated parent anion (mass 112 u) was not observed in the experiment (see text).

Mass.	Channel	ΔE_theo_	X_onset_		X_max_	
1st	2nd	3rd
113	C_4_H_3_NO_3_^−^	–0.98	0.0	0.0		
112	C_4_H_2_NO_3_^−^ + H	2.38	–			
67	C_4_H_3_O^−^ + NO_2_	1.44	2.3	3.1	3.5	
55	C_3_H_3_O^−^ + NO + CO	0.26	1.1	1.5	3.1	
46	NO_2_^−^ + C_4_H_3_O	0.93	1.1	1.5	3.1	3.7
42	C_2_H_2_O^−^ + NO + C_2_HO	2.50	2.4	3.7	4.1	8.7
41	C_2_HO^−^ + C_2_H_2_NO_2_	2.86	2.9	3.9	4.4	8.7
	C_2_HO^−^ + NO + CO + CH_2_	3.01				
27	C_2_H_3_^−^ + NO + 2 CO	0.77	2.3	3.3	3.7	
26	C_2_H_2_^−^ + NO + CO + COH	3.63	2.5	3.5		
	CN^−^ + C_3_H_3_O_3_	−3.02				
25	C_2_H^−^ + NO + CO + COH_2_	1.34	2.4	3.7		
16	O^−^ + C_4_H_3_NO_2_	2.23	3.1	4.3	9.1

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
