# Peer review of "Electron Attachment Studies with the Potential Radiosensitizer 2-Nitrofuran"

_ijms, 2020, doi:10.3390/ijms21238906_

Round 1

Reviewer 1 Report

This manuscript reports experimental and computational results on electron attachment with 2-Nitrofuran. The latter has been proposed as a radio-sensitizer and so its interactions with electrons of low energies, similar those of the secondary electrons generated in condensed matter by ionizing radiation, is a very suitable topic of study. The authors report total anion yields and the yields (and cross sections) of fragment anions generated by dissociative electro attachment, which they correlate to calculated energies of anion states. Possible dissociation pathways and dissociation dynamics are proposed and supported by further calculation.

The authors have considerable experience with these experimental and computational methods and studies of this type. The study is well-performed, appears very complete and should be published following minor revision. These changes relate exclusively to minor problems with English. I have listed an incomplete list of possible problems below, but the manuscript would benefit from proof reading by a native English speaker (or similar).

Some possible changes to the text.

Line 24: Together with chemotherapy and surgery, radiotherapy remains one of the most common and efficacious techniques for cancer treatment; globally, about 50% of patients receive radiotherapy as part of their treatment [1].  

L 169: with almost twice the signal of the parent anion. The formation of the complementary anion C4H3O– occurred was much weaker; other fragment anions are formed with similar intensities. In the original spectrum, it was not possible to resolve mass peaks in the range of m/z 25–28.  An improved measurement is shown as an inset to Figure 1, and resolves consecutive anionic fragments of similar intensity at m/z 25, 26, and 27.

L175: Negative ion mass spectrum of 2-nitrofuran upon electron attachment. The spectrum was obtained by summing 11 individual spectra recorded at electron energies from 0 to 10 eV, in steps of 1 eV. The inset shows anionic fragments formed in the range of m/z 25-28  with improved mass resolution. A schematic representation of the molecule is also shown.

L180: The energies of peak maxima (XMAX­) reported in Table 1, were determined by Gaussian fits.

L225: The absence of decomposition reactions in the low-energy region agrees with the calculated energetics of  dissociation reactions (Table 1) that are all endothermic except for the CN– channel which requires considerable molecular restructuring, as discussed below. In other words, there is no open dissociation channel for anions created at ~0 eV.

L234: Replace ‘By that process’ with ‘Accordingly’

L236: In electron attachment experiments with furan (C4H4O)

L239: Similar to 2-NIF, the addition of a nitrogen dioxide group also had a stabilizing effect on formation…

L256: which is underlined by the anion efficiency curve of the far less abundant C4H3O– (67 u) displayed in Figure 2c.

Author Response

See uploaded file

Reviewer 2 Report

The manuscript of Saqib et al. analyses the effects of low energy electrons and their ability to attach to 2-nitrofuran, a potential radiosensitiser. The manuscript is well written and follows previous studies by the authors along these lines, uncovering potential radio sensitisers, and is timely given the importance of the topic. I do have a few small comments I think need to be addressed prior to publication: 

  • In the computational details it is mentioned an active space of 3 electrons in 6 orbitals and a 6-31G* basis set as any more would be “beyond computational feasibility”. I find this statement to be simply wrong: there have been simulations with multiconfigurational methods on DEAs with much larger active spaces as far back as 2012(doi.org/10.1021/ct300153f), more complex approaches have been used to not only consider electron attachment but also electronic resonances (doi.org/10.1021/acs.jpca.8b01523), and even for the simulations with CCSD there are alternatives to the methods used available in a variety of software packages that are better built to treat anions, particularly dipole-bound ones (doi.org/10.1146/annurev-physchem-052516-050622) than the ones used in the present study. The authors should better justify why use such small spaces and basis sets, which lead to less electron correlated simulations, as this may well play a role in explaining anionic decay (see below comment related to Figure 5). Moreover, the authors should include a figure in the supporting information with the active space employed, as there is no unique way to build it and depending on the orbitals included the electron correlation retained might be different.

  • In line 171-173: It is mentioned how an improved measurement was carried out but no details are given as to how this was done; for someone not familiar with such experimental techniques it begs the question: if you can do an “improved” measurement, why not always do it at the highest quality? or is this meant to be just a zoom-in on that particular data range? A few more details here would help as the way it is written is confusing.

  • In Figure 4 caption, lines 214-217: the transition energies depicted are shifted by -0.6 eV and then -0.18 eV, but no detail how: does this mean all states are shifted -0.78 eV? or are some of them shifted by -0.6 and others -0.18, and if so which ones are which? It is also not clear what the values themselves refer to: does the -0.6 eV refer to the actual vertical electron affinity at CCSD? Because if this is the case it is a metastable anion, and without proper methods (i.e. stabilisation or a more diffuse basis) it is unlikely this will be accurate enough to provide a reliable reference to justify the -0.6 eV energy shift. For the -0.18 eV shift, why is the difference in energies averaged over the first 25 excited states if you are then using only 7-8? What would be the shift if it were averaged over those specific states instead, which are the only ones that are actually relevant?

  • Fig 5 depicts the reaction pathway to explain the decay of the electronic excited anionic state to the anionic ground state. It is not clear how this is obtained; the only optimised points are the starting FC point and the excited state minimum, but an interpolation is made to a pseudo conical intersection (not optimised, if it were energies would be degenerate to meV accuracy). The interpolation is interpreted as facilitating access to the conical intersection despite presenting a sizeable energy barrier (>1 eV) along the decay. Admittedly linear interpolations do provide upper bounds of potential energy barriers and the addition of correlation with MRCI reduces it, but this is an aspect that should be discussed as it is not obvious from looking at the profile. The role of the conical intersection is not really discussed; a minimum energy conical intersection optimisation (i.e. characterising its branching space) would provide information on whether the intersection is single path, or if it can both return to the FC or lead to a photo-product, which would be the fully dissociated NO2- species (+ its parent neutral species). Moreover, there is no information given on the relative energies of this fully dissociated NO2- species, which would complete this Figure.

  • Lines 362-363: a point is made about how the prevalence of NO2- species might be related to the barrier less decay and efficient passage through the conical intersection; optimising this structure and adding the detached (NO2- separated) structure to complete the potential energy surface for this process as suggested above would help explain this, which is one of the key points made in the manuscript to understand what is one of the main channels triggered upon LEE exposure.

  • In lines 375-379 an analogy between Furan and 2-NIF is drawn: Furan possesses an ionisation potential (IP) of 8.88 eV and features an ion yield signal between 10-11 eV, which was thus above the IP and therefore attributed potentially to Rydberg excited states of the anion. 2-NIF, on the other hand, has an IP of 10.04 eV and presents an ion yield peak at around 9 eV, i.e. below the IP, but it is also attributed to potential Rydberg excited states. I find this deeply confusing, can the authors further elaborate on this? If the ion yield being above or below the IP bears no relevance to the potential mechanism, then why mention it?

  • Not sure whether lines 388-394 should enter in the conclusions section, as they have nothing to do with the actual work carried out and its conclusions; they are background that could be included in the introduction where appropriate. 

Author Response

See uploaded file
